# Selective breeding enhances coral heat tolerance to marine heatwaves

Adriana Humanes [1,8] ✉, Liam Lachs [1,8], Elizabeth Beauchamp [1], Leah Bukurou[2], Daisy Buzzoni [3], John Bythell[1], Jamie R. K. Craggs[4], Ruben de la Torre Cerro[1], Alasdair J. Edwards[1], Yimnang Golbuu [5], Helios M. Martinez[1], Pawel Palmowski[1], Eveline van der Steeg[1], Michael Sweet [6], Alex Ward [1], Alastair J. Wilson [7] & James R. Guest [1]

Marine heatwaves are becoming more frequent, widespread and severe, causing mass coral bleaching and mortality. Natural adaptation may be insufficient to keep pace with climate warming, leading to calls for selective breeding interventions to enhance the ability of corals to survive such heatwaves, i.e., their heat tolerance. However, the heritability of this trait–a prerequisite for such approaches–remains unknown. We show that selecting parent colonies for high rather than low heat tolerance increased the tolerance of adult offspring (3–4-year-olds). This result held for the response to both 1-week +3.5 °C and 1-month +2.5 °C simulated marine heatwaves. In each case, narrow-sense heritability ($h^2$) estimates are between 0.2 and 0.3, demonstrating a substantial genetic basis of heat tolerance. The phenotypic variability identified in this population could theoretically be leveraged to enhance heat tolerance by up to 1 °C-week within one generation. Concerningly, selective breeding for short-stress tolerance did not improve the ability of offspring to survive the long heat stress exposure. With no genetic correlation detected, these traits may be subject to independent genetic controls. Our finding on the heritability of coral heat tolerance indicates that selective breeding could be a viable tool to improve population resilience. Yet, the moderate levels of enhancement we found suggest that the effectiveness of such interventions also demands urgent climate action.

The ability of wild populations to adapt to anticipated rates of climate change remains uncertain[1]. Corals are at the forefront of climate change impacts due to their vulnerability to marine heatwaves that lead to mass coral bleaching and mortality[2,3]. To persist, corals must adapt to ocean warming and the increasing intensity of heatwaves. This requires both genetic diversity and heritability of heat tolerance. Coral heat tolerance varies within populations[4],

between populations[5,6], across environmental gradients[7] and among taxa[8]. However, there are no reliable estimates for the heritability of heat tolerance that are relevant to both adult coral survival and the severity and duration of marine heatwaves.

Assisted evolution has been proposed as a strategy to mitigate coral reef degradation by facilitating adaptation in the face of climate change[9,10]. Such interventions may include modifications to host

[1]School of Natural and Environmental Sciences, Newcastle University, Newcastle upon Tyne, UK. [2]Palau International Coral Reef Center, Koror, Palau. [3]University of Victoria, Victoria, BC, Canada. [4]Horniman Museum and Gardens, London, UK. [5]The Nature Conservancy Micronesia and Polynesia, Koror, Palau. [6]Aquatic Research Facility, Nature-based Solutions Research Centre, University of Derby, Derby, UK. [7]Centre for Ecology & Conservation, University of Exeter, Penryn, UK. [8]These authors contributed equally: Adriana Humanes, Liam Lachs. ✉e-mail: adrihumanes@gmail.com

genetics, epigenetics, microalgal symbionts, or other microbial symbionts of the coral holobiont. One approach is to selectively breed corals for specific traits such as heat tolerance[11]. Selective breeding for stress tolerance has been successfully implemented in a wide range of organisms (e.g.,[12–16]), leveraging a variety of mechanisms. Recently, selective breeding for climate change adaptation has been trialled for economically important species with beneficial outcomes (e.g., drought tolerance in wheat[17,18], cold tolerance in spruce[15]), yet numerous challenges remain before such approaches can be implemented in wild populations (e.g., for amphibians[19], coral reefs. [20,21]). Trait heritability partly determines responses to selection, however, heritability is highly context-dependent, varying with life history stage, over time, and between populations[22]. Many stress tolerance traits are complex (i.e., controlled by numerous genes) and multivariate (e.g., for coral heat tolerance, the trait can be measured in various ways such as change in colour or photochemical efficiency). Furthermore, the heritability of stress tolerance traits has been shown to depend on the magnitude and duration of stress exposures[23]. The extent of trait enhancement possible from selective breeding is directly dependent on the narrow-sense heritability ($h^2$) of the target trait, defined as the proportion of phenotypic variance explained by additive genetic variance. This value is critical for estimating generational responses to selective breeding (c.f., breeder's equation[24]). Despite calls for selective breeding interventions aimed at boosting the tolerance of corals to marine heatwaves, the heritability of this trait remains uncertain.

Here, we test the viability of selectively breeding corals for heat tolerance (Fig. 1a) under conditions that approximate marine heatwaves (e.g., heat stress intensity and duration, bleaching and mortality responses) in a wild population of a common reef-building coral, *Acropora digitifera* (Fig. 1b). Current estimates of the additive genetic component of coral heat tolerance[25] are based on survival of early life stages[26–32] or bleaching responses of adults to heat stress exposures without taking their survival into account[33,34]. Remarkably, no study to date has estimated $h^2$ for any trait in selectively bred adult corals, presumably because of the difficulty and long time periods required to rear offspring corals to adulthood (multiple years). This is a critical knowledge gap because $h^2$ can vary across life stages and the greatest ecological impact from mass bleaching events is due to the loss of adult corals[35]. Here, we conducted selective breeding crosses among parents with low heat tolerance ($N = 7$) and among parents with high heat tolerance ($N = 7$, Fig. 1a), reared offspring until reproductively mature (i.e., 3-years-old, Fig. 1c) and assessed offspring heat tolerance using temperature stress exposures equivalent to those used for selecting parent colonies (Fig. 1d). We conducted two experimental trials to selectively breed either for tolerance to a short-term 1-week +3.5 °C stress, or a long-term 1-month +2.5 °C stress (Fig. 1d), with the latter being more typical of marine heatwave conditions[36]. This allowed us to estimate the genetic correlation between both traits, and so determine whether selecting on short-term heat stress tolerance is expected to also enhance long-term stress tolerance. This matters because short-term stress tolerance is easier to assay in practice, and so select upon, but may be less critical to the survival of corals in nature. As the genetic identity of the Symbiodiniaceae community can strongly influence the heat tolerance of the coral holobiont[37], we also tested for this effect. By using a realistic simulation of a marine heatwave and expressing heat tolerance in relation to mortality, we have ensured that this assessment of $h^2$ for coral heat tolerance is as ecologically relevant as possible.

## Results and discussion
### Characterisation of coral heat tolerance
While the loss of symbiotic microalgae (i.e., bleaching) is often used to measure coral heat tolerance, this does not always lead to mortality[38,39]. Therefore, we define coral heat tolerance as the combined bleaching and mortality response to a given heat stress dose. In coral reef ecology, the severity of mass bleaching events are typically predicted based on accumulated heat stress rather than lethal upper temperature limits. The temperature stress threshold for corals is theorised to be 1 °C above typical warm season temperatures (see MMM in methodology). Accumulated heat stress is then estimated in terms of degree heating weeks (DHW), which reflects both the duration and intensity of heat exposure above this threshold[40]. Significant reef-wide bleaching is expected at 4 °C-weeks, and widespread mortality at 8 °C-weeks[40]. In the short-term heat stress experiments that we conducted, temperatures were maintained at approximately 3.5 °C above the temperature stress threshold for approximately one week, leading to accumulated heat stress levels of 4 °C-weeks (Supplementary Table 1). In contrast, during the long heat stress experiments,

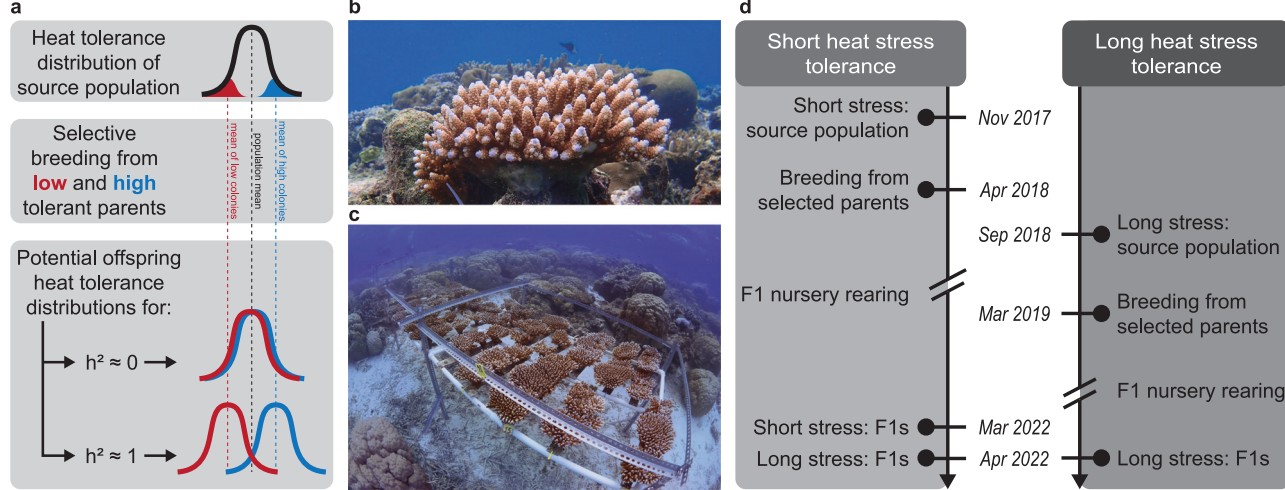

**Fig. 1 | Conceptual diagrams and experimental design. a** Effectiveness of selective breeding in shifting heat tolerance distributions (bell curves) of offspring from parents with low (red) or high (blue) heat tolerance is dependent on narrow-sense heritability ($h^2$). Dashed lines represent the mean of low heat tolerant parental colonies (red), the population (black) and high tolerant colonies (blue). **b** Corals in the source population were located on the reef and (**c**) selectively bred offspring (F1) colonies were reared in common garden nurseries. **d** Timelines are given for short- (left) and long-term (right) heat stress tolerance, showing dates of parental heat stress selection assays, selective breeding, nursery rearing, and adult offspring heat stress assays. Parental colonies were obtained from the same source population for both selective breeding efforts (2018 and 2019).

**a** Short-stress tolerance    **b** Long-stress tolerance

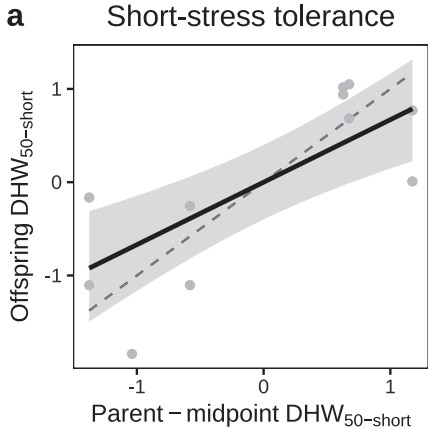

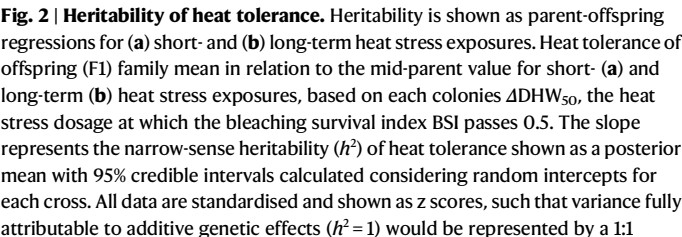

**Fig. 2 | Heritability of heat tolerance.** Heritability is shown as parent-offspring regressions for (**a**) short- and (**b**) long-term heat stress exposures. Heat tolerance of offspring (F1) family mean in relation to the mid-parent value for short- (**a**) and long-term (**b**) heat stress exposures, based on each colonies $\Delta DHW_{50}$, the heat stress dosage at which the bleaching survival index BSI passes 0.5. The slope represents the narrow-sense heritability ($h^2$) of heat tolerance shown as a posterior mean with 95% credible intervals calculated considering random intercepts for each cross. All data are standardised and shown as z scores, such that variance fully attributable to additive genetic effects ($h^2 = 1$) would be represented by a 1:1 relationship (dashed line) between the parent and offspring heat tolerances. The narrow-sense heritability ($h^2$) of short- and long-stress heat tolerance was 0.29 (±0.16 SE) and 0.23 (±0.16 SE), respectively, based on a frequentist animal model. This significant heritability is corroborated by the parent-offspring regressions presented here with higher but overlapping $h^2$ estimates compared to the uncertainty of the animal model for short- and long-stress heat tolerance of 0.67 (95% credible interval: 0.34−0.98) and 0.52 (95% credible interval: 0.22−0.83), respectively. The predicted regression (bold line) and Bayesian 95% credible intervals (shading) are shown. Source data are provided as a Source Data file.

temperatures were maintained at approximately 2.5 °C above the stress threshold for over 30-days, leading to accumulated stress levels exceeding 10 °C-weeks (Supplementary Table 1). The latter long-term heat stress assay was designed to simulate natural marine heatwave conditions[4,41]. For example, during the 2016 bleaching event on the Great Barrier Reef, approximately 30% mortality was initially documented at a reef-wide scale after 8 °C-weeks[42], and mass bleaching occurred in Palau (the location of this study) in 1998 and 2010 from marine heatwaves reaching 7-8 °C-weeks[43,44].

### Adult coral heat tolerance is heritable

We found considerable variability of heat tolerance within the study population of *Acropora digitifera*, allowing us to selectively breed parental colonies for low and high heat tolerance and estimate $h^2$ from offspring phenotypes. Using a bivariate frequentist animal model with pedigree-based relatedness among individuals, we show that short- and long-term heat stress tolerance are both heritable, with $h^2$ estimated as 0.29 (±0.16 SE) and 0.23 (±0.16 SE), respectively. This was corroborated using parent-offspring regressions (Fig. 2), with $h^2$ estimates of short- and long-term stress tolerance (estimated at 0.67 (95% CI: 0.34−0.98) and 0.52 (95% CI: 0.22−0.83), respectively) overlapping with the uncertainty limits of the animal model. These higher values are consistent with a general pattern in the literature that parent-offspring regressions are prone to sources of upward $h^2$ bias[45,46]. These results collectively demonstrate a considerable host genetic contribution to variation in coral heat tolerance (although with some uncertainty in the precise value of $h^2$ reflected by relatively wide SEs) and suggest scope for both natural adaptation and assisted evolution. Additional single parent offspring regressions were qualitatively suggestive of steeper dam-offspring slopes relative to sire-offspring effects (Supplementary Fig. 3). Although the differences were not significant, if confirmed it may suggest some involvement of maternal effects or other non-additive forms of inheritance.

### Selection enhances coral heat tolerance

For the first time, we have shown that a single generation of selection (Fig. 3a−d) can produce a significant change in the heat tolerance of adult offspring (Fig. 3e−h). This was true both when selecting on short- (Fig. 3e, g) and long-term heat stress tolerance (Fig. 3f, h). Having high heat tolerant parents provided enhanced heat stress resistance to offspring, with significant shifts in heat tolerance distributions between offspring from low-low (LL) families (low sire × low dam) compared to those from high-high (HH) families (high sire × high dam) (Fig. 3e, f). This significant difference in heat tolerance between LL and HH families also held true for the progression of bleaching and mortality responses (BSI – bleaching survival index) throughout the stress experiments (Fig. 3g, h). As a result, the HH families could withstand approximately 1 °C-weeks additional heat stress compared to the LL families (0.8 and 0.9 °C-weeks for short- and long-stress tolerance, respectively). Logistical constraints and time needed to rear selected lines of corals to reproductive maturity prevented us from conducting population and generational replication of high and low selected lines. This means that we cannot exclude genetic drift as a driver of the observed trends (i.e., the change in allele frequencies between the wild population and the selectively bred offspring cohort due to random chance). However, taken together our results (Figs. 2, 3) indicate that heritability of heat tolerance is facilitating the selection response.

Previous research has identified substantial variability in heat tolerance for *A. digitifera* in Palau[4,47]. Here, we did not breed from the most heat tolerant individuals identified in this population. Therefore, there is potential to apply stronger selection; if selective breeding efforts could find the top 1% most-tolerant corals in the population (which would be 3 °C-weeks more tolerant than the population mean based on our results), then first-generation offspring would likely be able to withstand additional heat stresses of approximately 1 °C-week. Additional gains could be achieved using multiple rounds of selection over numerous generations and by selecting for other adaptive traits (e.g., shorter generation times). Despite all of this, potential enhancements could still be modest compared to projected ocean warming over the coming decades[4,48].

### No effect of symbiotic microalgae

While our analysis indicates that host genes have a strong influence on heat tolerance (also see ref. 34), many other drivers are also known, including thermal history[49], Symbiodiniaceae community[37], and other components of the microbiome[50]. We were able to investigate the Symbiodiniaceae community but found no evidence of effects on heat tolerance. *Cladocopium* spp. symbionts dominated 95% of coral

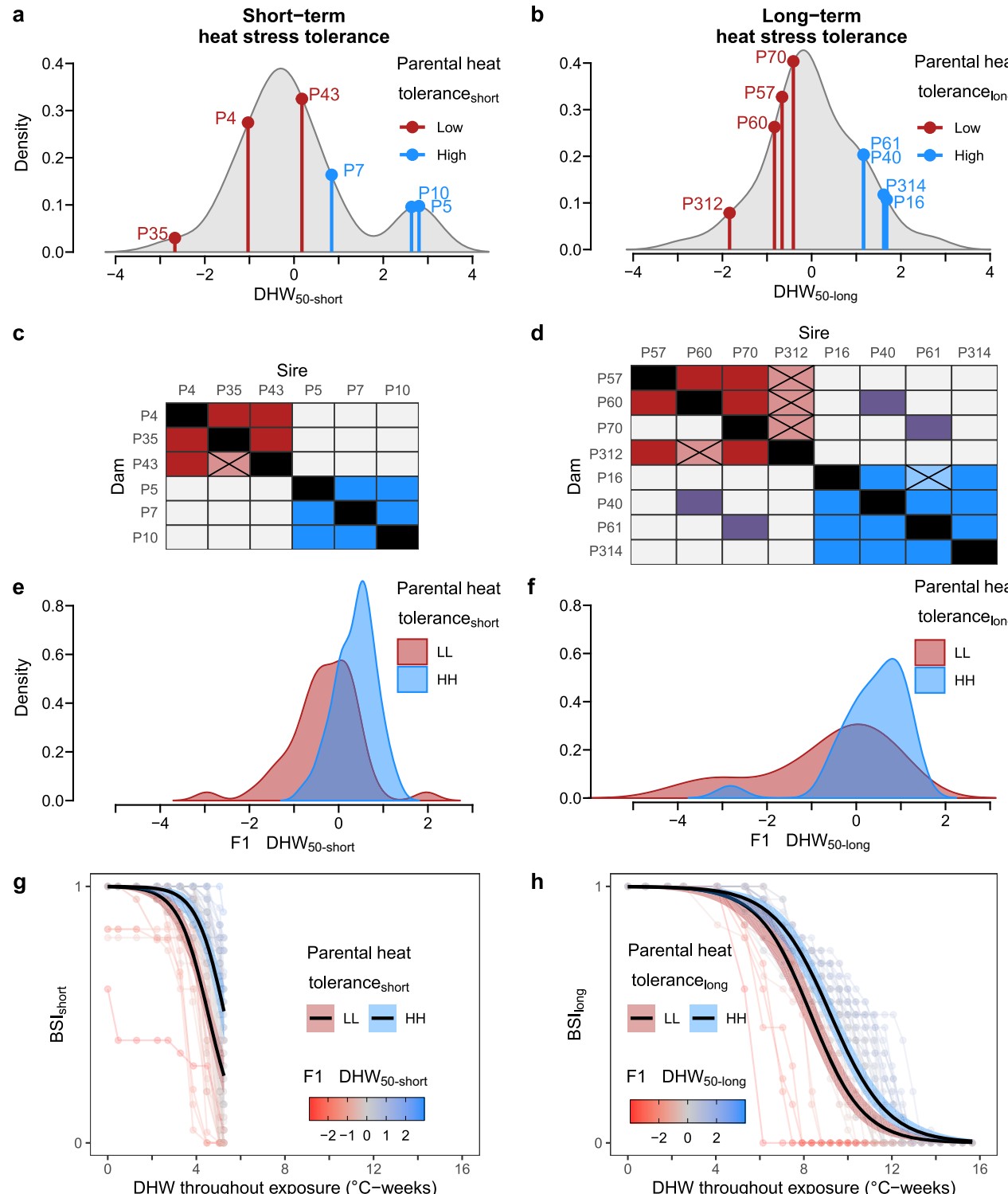

**Fig. 3 | Response to selective breeding for heat tolerance.** Selection for short- (**a**, **c**, **e**, **g**) and long-term (**b**, **d**, **f**, **h**) heat stress tolerance comparing parents and adult offspring (F1). Distributions of heat tolerance (kernel density of $\Delta DHW_{50}$, the heat stress dosage at which a colony's BSI passes 0.5) in the source population to short- (**a**) and long-term (**b**) heat stress exposures ($n = 31$ and $n = 65$ colonies exposed to short- and long-term stress, respectively). Heat tolerance of selected parent colonies ('P' in the colony ID) is indicated by vertical lines for low (red) and high (blue) heat tolerance. **c**, **d** Matrices show crosses conducted, either low-low (LL, red), high-high (HH, blue), or low-high (purple), highlighting those from which between low- (red) and high-tolerant (blue) parents, based on short- (no individuals survived until offspring heat stress (X). **e**, **f** Offspring heat tolerance distributions between LL (red) and HH (blue) are significantly different based on Linear Mixed Models (**e**, $z = 5.1_{88/84}$, $P = 3 \times 10^{-7}$; **f**, $z = 2.7_{43/39}$, $P = 0.008$). **g**, **h** Progression of instantaneous BSI for each offspring colony (faint lines and points coloured by F1 $DHW_{50}$) throughout the heat stress exposure, in terms of DHW. The overall high- or low-selected sigmoidal BSI-DHW responses (black lines and shaded 95% confidence intervals) show significantly different intercepts between LL and HH crosses based on Generalised Linear Mixed Models (**g**, $z = 5.3_{880/875}$, $P = 1 \times 10^{-7}$; **h**, $z = 3_{1280/1275}$, $P = 0.002$). Source data are provided as a Source Data file.

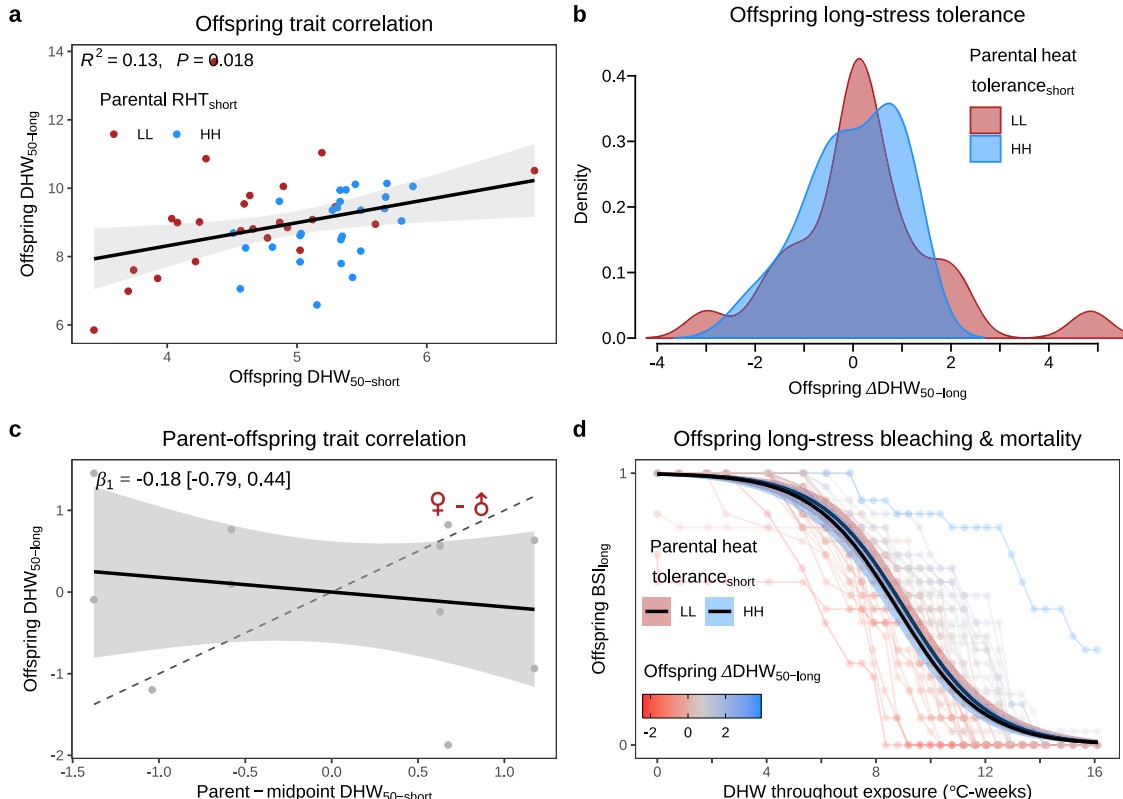

**Fig. 4 | Phenotypic and genetic relationships between short- and long-term heat stress tolerance.** Offspring selected for short-term heat stress tolerance (F1-2018) and subsequently subjected to both short- and long-term heat stress exposures. **a** Phenotypic trait correlation between short- and long-stress tolerance for the F1 cohort (in terms of $DHW_{50}$), showing a significant positive correlation between these two traits (Linear Mixed Model fit shown on plot with 95% confidence intervals, $Z = 2.4$, and bivariate animal model residual trait correlation, $r_R = 0.48$ (0.18 SE), $Z = 2.6_{50/42}$, $P > 0.05$). **b** Selection for short-stress tolerance did not yield changes in long-stress tolerance. This is shown as distributions of offspring long-stress tolerance, grouped by their parents short-stress tolerance (low or high), with no significant difference in $\Delta DHW_{50\text{-long}}$ between LL and HH families based on Generalised Linear Mixed Model ($Z = -0.6_{50/46}$, $P > 0.05$). **c** Relationship between parent midpoint short-stress tolerance and their offspring's long-stress tolerance based on z-scored data (points). The linear regression slope ($\beta_1$, the mean

and 95% credible intervals of slope posterior distribution), and the predicted regression (mean and 95% credible intervals) show the trend is no different than zero (i.e., a flat line). A 1:1 relationship (dashed line) would suggest that the same genetic controls are present for both traits. The lack of a significant genetic correlation was also shown from a bivariate animal model ($r_G = 0.06$ (0.66 SE), $Z = 0.09$, $P = 0.90$). **d** Progression of BSI for each F1 offspring (faint lines and points) throughout the long-term heat stress exposure (degree heating weeks – DHW) corroborated the non-significant genetic correlation between short- and long-stress tolerance. Cohort-level sigmoidal dose-response curves are grouped by parental short-term heat stress tolerance showing no significant difference between LL and HH families based on Generalised Linear Mixed Model ($Z = -0.7_{1500/1495}$, $P > 0.05$, error bands are 95% confidence intervals). Source data are provided as a Source Data file.

colonies studied (7 hosted mainly *Durusdinium* and 2 were dominated by *Symbiodinium*). Physiological traits such as heat tolerance can vary extensively between species within the genus *Cladocopium*[51], but ITS2 (Internal Transcribed Spacer) profiling revealed no significant differences in coral heat tolerance between those hosting different Symbiodiniaceae genera (Supplementary Fig. 4.) or putative taxa (ITS2 type profiles, Supplementary Fig. 5). As such, differences in Symbiodiniaceae assemblages were unlikely to be major drivers of coral heat tolerance variation in our study, supporting previous findings from the same source population[4,47].

## Traits with potential independent genetic control

Given the impacts of climate change on coral reefs globally, it is imperative to develop heat tolerance assays that can be deployed rapidly, both to improve understanding of natural variability (within and between populations) and to identify colonies for assisted evolution and restoration. Previous studies have suggested that the response of coral holobionts to rapid thermal challenges (hours) is predictive of their response to moderate-term exposures (i.e., 21 days[52]). Supporting this, our data show that there is a weak, but significantly positive, phenotypic correlation between a colony's

response to short- versus long-term heat stress exposures (Fig. 4a). However, we find no support for a genetic correlation between these traits from a bivariate animal model, $r_G = 0.06$ (±0.66 SE, $P > 0.05$, Fig. 4c). Further, corals selectively bred for short-stress tolerance did not show enhanced long-stress tolerance (Fig. 4b, d). Together, our findings suggest that these two forms of heat tolerance (short vs. long) are both heritable (Fig. 2) but, importantly, may be under independent genetic controls as seen in other systems[13]. Clearly our estimate of $r_G$ is highly uncertain and this latter conclusion remains tentative. However, the result highlights an important consideration: that selecting for a heat tolerance trait that is genetically uncorrelated (or even negatively correlated) with the ability to survive marine heatwaves could waste both limited resources and the short time window remaining to facilitate climate adaptation. It remains imperative to better understand the genetic controls of different forms of heat tolerance and their genetic correlations to accurately predict the response of corals to marine heatwaves under climate change.

## Implications

In order for corals to adapt to climate warming and for assisted evolution interventions to be effective, adult coral heat tolerance must be

heritable. For the first time, our study shows that this trait is heritable and so amenable to enhancement through natural selection and selective breeding. While mass bleaching mortality events can be used to identify heat tolerant genotypes for breeding (i.e., those colonies that survive), this requires waiting for destructive events to occur, highlighting the need for alternative methods to pre-emptively select tolerant colonies. Compared to long-term stress tests, rapid assays offer clear logistical advantages. However, we caution that the positive phenotypic correlation between short- and long-stress tolerance is not demonstrably underpinned by a genetic correlation, highlighting this as a critical future research priority. Selection of brood stock for short-stress tolerance did not provide adult offspring with enhanced tolerance to simulated long-term marine heatwave conditions. Indeed, care must be taken when using short-term laboratory assays to investigate the underlying mechanism of tolerance for corals that survive natural mass bleaching events. Further research is needed to uncover the genetic covariances between fitness-related traits (i.e., tolerance to short- versus long-term stress exposures), and to upscale selective breeding for management interventions. Ultimately, accurate low-cost, rapid assays need to be developed to enable robust selection of required heat tolerance traits for assisted evolution to be implemented successfully.

The proposition of using assisted evolution interventions to bolster reef resilience in the face of climate change is still highly controversial. It remains unknown whether such novel approaches will provide sufficient ecological benefits to warrant the expense and effort required to implement them. The results of our research simultaneously offer reasons for both hope and caution. On one hand, we found heat tolerance to be sufficiently heritable to suggest selective breeding could lead to significant enhancements in offspring population heat tolerance. On the other, the response to selection was relatively modest compared to projected ocean warming for the coming decades. In practice, to achieve significant trait enhancements that can keep pace with climate change will require ongoing rounds of selection over multiple generations – as is a standard approach for selective breeding programmes in other organisms[53] – and will require us to overcome several remaining challenges regarding how to incorporate selective breeding into reef rehabilitation programmes. These include targeting the appropriate species to benefit broader reef communities, upscaling selective breeding and out-planting efforts, overcoming post-deployment survival bottlenecks, and integrating with stakeholder and governance frameworks whilst securing sustainable financing models. In summary, our results show sufficient promise to warrant further research and development into selective breeding as an assisted evolution tool. However, we urge caution using these approaches until we have a better understanding of the relative benefits and risks. Without question, such efforts are only going to be worthwhile if in the long term we secure a future for coral reefs by rapidly reducing global carbon emissions and managing local-scale human disturbances. These remain the greatest priorities for coral reefs and the people who depend on them.

## Methods

### Parent colony selection using heat stress experiments

All efforts were made to collect and export samples in compliance with local, national and international laws. Both national and state permits were obtained prior to any work commencing and all work was done with full collaboration of the Palau International Coral Reef Center. This work was conducted using Koror State permits (018, 032, 034, 037), Palau National permits (RE-18-13, RE-19-08), and CITES export permits (permit number PW19-111). The reef-building coral *Acropora digitifera* was used as a model species given its widespread distribution and abundance on shallow reefs throughout the western Indo-Pacific. All corals were sourced from an outer reef crest sharing the same thermal history in the Republic of Palau (Mascherchur,

N 07°17′ 29.3″; E 134°31′ 8.0″), where *A. digitifera* is abundant at depths ranging between 0.5 and 4 m. The heat stress experiments to select the parent colonies for brood stock were conducted at the Palau International Coral Reef Center (PICRC) in 2017 and 2018 and are described in detail in refs. [21,47]. Briefly, fragments collected from 34 tagged visibly healthy colonies in November 2017 were exposed to a short-term[53] 7-day heat stress assay, after a sufficient healing/acclimation period (see refs. [53,54], Supplementary Table 1). Fragments were randomly distributed among five heat stress tanks and three procedural control tanks, ensuring that each colony had at least one fragment in a control tank. Temperatures in stress tanks were raised incrementally over the course of three days (+2 °C on day one, and +1.5 °C on day three), reaching a daily average temperature of 33.0 °C (±0.37, Supplementary Table 1), whereas control tanks remained at ambient seawater temperatures (30.4 ± 0.5 °C, Supplementary Table 1). In June 2018 the same procedure conducted in November 2017 was used to expose fragments from 66 colonies, but this time fragments were exposed to a long-term[53] 35-day stress assay. For this long-term heat stress exposure, fragments were randomly distributed among four heat stress tanks and two procedural control tanks, ensuring that each colony had at least one fragment in a control tank and the remaining in independent stress tanks. Temperatures in stress tanks were gradually raised from ~29.5 °C to ~32.8 °C (+0.5 °C on days 1, 9, 18, 22, 28, 29 and 32, Supplementary Table 1), whereas control tanks remained at ambient seawater temperatures (29.3 ± 0.7 °C, Supplementary Table 1).

The status of each fragment was visually inspected by the same observer at intervals of one to five days (depending on the changes in their pigmentation) and ranked as: (0) healthy (no signs of discolouration or mortality), (1) partially bleached (<50% of the tissue was bleached), (2) fully bleached (>50% of the tissue was bleached), (3) partially dead (loss of some live coral tissue) or, (4) dead (loss of all live coral tissue). Notably, the bleaching status scores used here (0, 1, and 2) have previously been shown to correspond strongly to the 'whiteness' of coral fragments (measured as the Euclidean distance to the red-green-blue channel values from standardised images of the fragments), chlorophyll and carotenoid concentrations, and the tissue population density of dinoflagellate algal symbionts (see supplementary materials of[4]).

### Parental heat tolerance categories

Parental heat tolerance categories (low and high) were determined by the survivorship of replicate fragments when the entire experimental population reached 50% mortality. Colonies that had all replicate stressed fragments alive at 50% population mortality were considered to have high tolerance, whereas colonies with all stressed fragments dead at this time were classified as having low tolerance. It must be noted that these categorisations are purely relative and pertain to the stress test conducted. Colonies that were not classified either as high or low tolerance were considered as 'unclassified'. To ensure quality of the data, we removed colonies from the experiment if: (1) the fragment in the control tank died at any stage of the data analysis as this could be indicative of handling effects for that colony (n = 4, 1, 2, and 0 colonies in 2017, 2022 short-term, and 2018, 2022 long-term, respectively), (2) fewer than two fragments were alive at the beginning of the experiment after any losses during the acclimation period (n = 1 colony in 2017 short-term, and 0 colonies in the other three experiments)

### Accumulated heat stress

Typically, mass coral bleaching is predicted based on the amount of accumulated heat stress, rather than a specific instantaneous temperature[40]. The National Oceanic and Atmospheric Administration's Coral Reef Watch (NOAA CRW) measure accumulated heat stress as DHW based on their daily global sea surface temperature (SST) dataset CoralTemp v3.1[40]. NOAA CRW provide a near real-time

bleaching risk forecast where DHW of 4–8 °C-weeks indicates that significant bleaching is to be expected, and DHW > 8 °C-weeks indicates further mortality is to be expected. For this daily satellite data product, NOAA CRW use the following methodology. Daily SST data are converted to temperature anomalies by subtracting the local maximum of monthly means climatology (MMM–the average SST of the hottest month for a given satellite data grid cell and reference baseline period), and then transformed to be positive-only HotSpots (replace negative values with zero)(1). DHW (2) on each day (i) is calculated as the sum of Hotspots > 1 °C during the previous 12 weeks (84 days inclusive) and divided by 7 to make a weekly metric.

$$HotSpot_i = SST_i - MMM, \ HotSpot_i \geq 0 \qquad (1)$$

$$DHW_i = \sum_{n=i-83}^{i} \left( \frac{HotSpot_i}{7} \right), \ for HotSpot_i \geq 1 \qquad (2)$$

Here we applied an amendment to this methodology in order for experimental tank-based DHW measurements to be comparable to the NOAA CRW bleaching alert system (see DHW thresholds above[4]). First, the local satellite-based climatological baseline (MMM) was adjusted based on the relationship between satellite-sensed SST and in situ water temperature recorded from loggers on the home reef (parental colonies) or nursery (offspring colonies). Second, to account for the higher temporal resolution of tank experiment HotSpots measured using calibrated temperature loggers (1-min recording interval for parental short stress experiment or 10-min recording interval for all other experiments), the appropriate division is used to achieve a weekly DHW metric.

## Continuous heat tolerance metrics

A bleaching survival index (BSI)(3) was used to estimate coral colony phenotypic responses to the heat stress[55]. Here $c_1$ to $c_5$ are the proportion of coral fragments that are in each health status category: healthy ($c_1$), partially bleached ($c_2$), fully bleached ($c_3$), partially dead ($c_4$) or dead ($c_5$). N denotes the number of health status categories.

$$BSI = 1 - \frac{0 \times c_1 + 1 \times c_2 + 2 \times c_3 + 3 \times c_4 + 4 \times c_5}{N - 1} \qquad (3)$$

As such, colony BSI values range between zero and one, where a value of zero is achieved if all replicate fragments for that colony are dead, and a value of one is achieved if all replicate fragments are alive and healthy. As the BSI of a particular colony is representative of only a single timepoint and will change throughout the heat stress exposure, a mean BSI value per colony was estimated by averaging daily BSI values during the stress exposures. Lower mean BSI values indicate a higher percentage of bleaching and mortality, and thus a low tolerance to heat stress exposure, whereas higher BSI values indicate higher tolerance to heat stress exposure.

To improve the interpretability of a single continuous heat tolerance value per colony (mean BSI) we computed the DHW at which BSI passes a value of 0.5 ($DHW_{50}$), analogous to the $EC_{50}$ metric used in toxicology which refers to the concentration required to elicit 50% of the response being measured[56]. Since three of the four experiments were stopped at 50% mortality (meaning that BSI did not drop below 0.5 for every colony), we estimated $DHW_{50}$ for each colony in each experiment. $DHW_{50}$ was calculated by first fitting a logistic dose response curve for each colony using a generalised linear model with a single slope value (BSI ~ DHW + Colony), and then computing the $DHW_{50}$ for each colony as the DHW at which its logistic curve passes a BMI of 0.5 using the 'qlogis' function in R. This led to extreme overestimation of $DHW_{50}$ for colonies with a mean BSI of 1 ($N = 3$, source population short-stress, occurring when all replicate nubbins stayed

healthy throughout the entire experiment). For these colonies, we re-estimated their $DHW_{50}$ from the log linear relationship between mean BSI and $DHW_{50}$ based on all other colonies in that experiment (Supplementary Fig. 2, $R^2 = 98\%$).

## Selective breeding, larval settlement and offspring rearing

In April 2018 and March 2019 reproductive and apparently healthy colonies ($N$ = six and eight respectively) with either high or low heat tolerance classifications (Fig. 2a, b) were collected a few days before the full moon of the expected spawning month (April 1, 2018 and March 21, 2019, respectively). The selection of colonies was based on their heat tolerance category (low or high) and whether they were gravid (i.e., whether they contained mature pigmented oocytes) at the time of collection. In 2018, some of the colonies with known heat tolerances (short-term heat stress 2017, $N = 31$) had already spawned at the time of collection, which limited the number of colonies available for selective crosses. In terms of $DHW_{50\text{-short}}$ (short-stress tolerance), the selected high and low parents were in the top 13% and bottom 23% of known colonies, respectively. However, the low tolerant parent (P43), was an exception, being in the 71st percentile, but was the only suitable gravid colony available to achieve the minimum required number of low parents for the short-stress selection experiment ($N = 3$). In terms of $DHW_{50\text{-long}}$ (long-stress tolerance), the selected high and low parents were in the top 11% and bottom 32% of known colonies, respectively. These logistical constraints, which may limit the selection response of breeding interventions, could be mitigated in future work with several easily implementable steps. (1) Tag and phenotype as many colonies as logistically possible to avoid problems with colonies dying, not being gravid, or not being found. In this study, we encountered these problems with 31 colonies, so recommend using many more, although this could be species and site specific. (2) Phenotyping could be conducted close to the time of spawning to reduce chances of colonies dying or being lost in between phenotyping and spawning. (3) If split spawning is a possibility, schedule breeding for both spawning events to maximise the chance of success. Cryopreservation of sperm could also be considered as a backup if too few colonies spawn[57].

In 2018 and 2019 two types of crosses were produced: high sire × high dam (HH), and low sire × low dam (LL) (Fig. 2C). Reciprocal crosses were also conducted in 2019: high sire × low dam (HL), and low sire × high dam (LH) (Fig. 2d). The use of reciprocal crosses (i.e., cross 1_2 = eggs from parent 1 and sperm from parent 2, and cross 2_1 = eggs from parent 2 and sperm from parent 1) means that potential maternal effects will not confound estimates of heritability. Spawning, fertilisation, larval rearing, settlement, and colony rearing were conducted ex situ at the Palau International Coral Reef Center. We employed standard methods for coral rearing[58] with modifications to ensure isolation of individual crosses. Starting at sunset (19:00), colonies were monitored for signs of bundle setting. Upon observing the first colony setting, all colonies were isolated in individual static tanks. Plastic cups were used to collect bundles from the water surface when most bundles were released. The egg-sperm bundles were then separated by transferring them onto a 100 μm mesh filter within a bowl containing a small amount of UV-treated (Trop UV Steriliser Type 6/IV – TPE, Trop-Electronic GMbH, Germany) 0.2 μm filtered sea water (FSW). Sperm remained in the bowl while eggs remained immersed in FSW within the filter. The filter was promptly transferred to a new bowl with fresh UV-treated FSW, and the eggs were rinsed five times to eliminate any remaining sperm. To prevent cross-contamination, all equipment including bowls, filters and utensils were rinsed with a diluted 1% bleach in FSW. All implements were labelled and dedicated to a specific colony or cross. cross-fertilisation was then performed to produce different crosses, which were subsequently reared in 15 L cone-shaped tanks at ambient temperature, with a flow rate of 0.2 L/min of UV-treated FSW, ensuring a full turnover every hour[21]. Gentle aeration was introduced 24 h post-fertilisation, once the embryos had developed

sufficiently to become round and motile. Larvae were offered circular ceramic substrates (Oceans Wonders LLC), overgrown with crustose coralline to facilitate settlement. Three days post-fertilisation, larvae were transferred from larval rearing tanks to settlement tanks with conditioned substrates. Two days later, the substrates were moved to flow-through nursery tanks (ex situ nurseries).

The resulting F1 offspring were tagged with a cable tie using a colour coded system to identify from which cross it had originated. Colonies were raised in ex situ nurseries for 13-months for the 2018-F1s and 6-months for the 2019-F1s before being transferred to in situ nurseries (N 7°18'19.80"N; E 134°30'6.70"E, Fig. 1b) 2.2 km away from Mascherchur reef. Colonies were raised in these nurseries until April 2022 (i.e., when they were 3 and 4 years old for the 2018 and 2019 F1 cohorts, respectively). At this time, colony sizes for the F1 generation (geometric mean diameter of $17 \pm 4$ SD and $10 \pm 3$ SD cm for 2018 and 2019 offspring, respectively, Supplementary Fig. 6, Fig. 1c) were equivalent to the minimum size of fecund adult colonies recorded from the wild population (13 cm diameter[47]). Fragments of F1 colonies from each cross were collected to conduct heat stress experiments to estimate their heat tolerances.

### Offspring heat stress experiments

In April 2022, six fragments per colony from 68 colonies corresponding to 11 unique crosses produced in 2018 (F1-2018, Fig. 1c) were exposed to a short-term heat stress (Supplementary Fig. 1c) of similar duration and intensity to the one used to select their parental colonies (Supplementary Fig. 1a, Supplementary Table 1). Similarly, in May 2022, six fragments from 54 colonies corresponding to 21 unique crosses produced in 2019 (F1-2019, Fig. 1d) were exposed to a long-term heat stress (Supplementary Fig. 1d, Supplementary Table 1) equivalent to the one used to select their parental colonies in 2018 (Supplementary Fig. 1b). We also aimed to test whether a colony's phenotypic response during a long-term heat stress exposure could be predicted by its phenotypic response to a short-term heat stress. To achieve this, six fragments from 50 of the F1-2018 colonies used in the short-term heat stress assay were also included in the long-term heat stress. The mean BSI and $DHW_{50}$ metrics of heat tolerance were estimated for each colony under each heat stress exposure.

### Symbiodiniaceae community

The symbiotic dinoflagellate (Symbiodiniaceae) community associated with each colony was assessed by internal transcribed spacer (ITS2) profiling. A ~ 2 cm fragment taken prior to the heat stress experiment was preserved in molecular grade ethanol and stored at −80°C prior to DNA extraction. Samples were air dried and DNA extracts ($n = 179$ samples from 61 colonies of the parental cohort, 65 colonies F1-2018 and 53 colonies F1-2019) were obtained using the Qiagen DNeasy blood and tissue kit following the manufacturers protocol, including overnight proteinase K digestion. Amplicon-based sequencing was performed on the Illumina MiSeq platform at the Genome Centre, Queen Mary University, London, UK. Briefly, the ITS2 primers SYM_VAR_5.8S2 (5'-<u>TCGTCGGCAGCGTCAGATGTGTATAAGA GACAGG</u>AATTGCAGAACTCCGTGAACC-3') and SYM_VAR_REV (5'-<u>GTC TCGTGGGCTCGGAGATGTGTATAAGAGACAG</u>CGGGTTCWCTTGTYT GACTTCATGC-3') (Illumina adaptors underlined) were used in a first-round PCR to amplify an approximately 450 bp fragment of the ribosomal ITS2 region[59]. Samples were indexed using Illumina Nextera DNA unique dual indexes and underwent quality control using Qubit 3.0 fluorometric quantification and size assessment using the Tapestation 4200 D1000. The pooled, equilibrated samples were run as paired end reads on an Illumina MiSeq v3 with an addition of a 20% PhiX control. Combined DNA extraction and PCR procedural controls were used as negative controls. ITS2 sequence data were submitted to SymPortal[60] to cluster defining intragenomic variants (DIVs) into profiles representing putative symbiont species. Some negative controls recovered

identifiable ITS2 profiles, and this was detected in one of our samples, which was removed from subsequent analyses. Relative abundance and sequence read data for DIVs and symbiont profiles are publicly available at www.symportal.org.

### Data analysis

Distributions of coral heat tolerance were analysed independently for each heat stress experiment and cohort (i.e., parental-2017 short-term, parental-2018 long-term, F1-2018 short-term, F1-2018 long-term, F1-2019 long-term). Heat tolerance as $DHW_{50}$ was either shown as a difference compared to the population mean value for that experiment ($\Delta DHW_{50}$), or as a z-score. Offspring heat tolerance distributions for HH and LL crosses were visualised as density kernels independently for the stress durations and offspring cohorts (i.e., F1-2018 short-term, F1-2018 long-term, F1-2019 long-term). The effect of parental heat tolerance (two-level fixed effect factor, HH vs. LL) on offspring $DHW_{50}$ (response) was tested using Linear Mixed-effect Models (LMMs) with a random intercept for each parent pair. The progression of colony BSI (response) throughout the heat stress exposure (DHW, fixed effect factor) depending on parental heat tolerance (two-level fixed effect, HH vs. LL) was tested using Generalised Linear Mixed-effect Models (GLMMs) with a binomial error distribution and random intercept for each colony. The difference in response DHW dosage between LL and HH offspring groups (i.e., the difference in DHW at which offspring groups pass BSI of 0.5, analogous to the colony-level $DHW_{50}$ metric) was calculated from model coefficients.

The $h^2$ of short-stress tolerance ($DHW_{50\text{-short}}$) and long-stress tolerance ($DHW_{50\text{-long}}$) were estimated using a bivariate animal model in ASReml-R ($N = 223$ colonies). The known pedigree of colonies from the breeding design was used to specify the additive genetic relatedness matrix which was then used together with parental phenotypes (31 and 65 colonies for $DHW_{50\text{-short}}$ and $DHW_{50\text{-long}}$, respectively) and offspring phenotypes (88 and 104 colonies for $DHW_{50\text{-short}}$ and $DHW_{50\text{-long}}$, respectively) to estimate the additive genetic variance for each trait as well as the genetic correlation between both traits ($r_G$). The statistical significance of $r_G$ was tested by comparison to an animal model fitted without genetic correlations using a $\chi^2$ test. To corroborate the animal model results, we also calculated $h^2$ from Bayesian parent-offspring regressions via Integrated Nested Laplace Approximation (INLA) based on the family mean of offspring following[61]. The parent-midpoint was used as the parental trait value such that the $h^2$ is estimated as the regression slope. A weakly informative prior estimate of the default Gaussian slope (i.e., $\beta_1$ or $h^2$) was constructed such that values outside the 0–1 range (the underlying prior assumption of $h^2$) were highly unlikely, achieved by setting the prior mean to 0.5 and the prior standard deviation ($\sigma$) to 0.255 (so that the mean $\pm 1.96 \times \sigma$ is 0–1). Note that the prior $\sigma$ is set in R-INLA using the precision parameter, where precision is the inverse of the variance ($1/\sigma^2$). An indication of maternal and paternal effects on offspring traits were also tested using mother-offspring-family-mean and father-offspring-family-mean regressions following the same approach[24,62] of constructing weakly informative priors. For all parent-offspring regressions trait values were z-scored prior to model fitting. The influence of the weakly informative $\beta_1$ prior was also tested by using default non-informative $\beta_1$ priors (prior mean and precision set to 0 and 0.001, respectively) and then rerunning parent-offspring regressions (Supplementary Fig. 7, equivalent result to Fig. 2), and mother- and father-offspring regressions (Supplementary Fig. 8, equivalent result to Supplementary Fig. 3).

The influence of the Symbiodiniaceae community on heat tolerance was tested at the genus level and at the putative taxa level (ITS2 profile). The effect of symbiont genera on $DHW_{50}$ was tested for each stress experiment individually (i.e., parental short-term, parental long-term, F1 short-term, F1 long-term) using non-parametric Kruskal-Wallis tests. The effect of individual ITS2 profiles on the intercept and slope of sigmoidal BSI-DHW regressions (BSI response variable, DHW × ITS2 profile

fixed effects) were tested for each stress experiment independently using GLMMs with a binomial error distribution, and a random intercept for each colony. Pairwise comparisons of ITS2 profile intercepts and ITS2 profile slopes were conducted using post-hoc Tukey tests.

The link between short-stress tolerance ($DHW_{50\text{-short}}$) and long-stress tolerance ($DHW_{50\text{-long}}$) was tested using four approaches. (1) The phenotypic correlation between the two traits was tested using a simple linear regression for F1-2018 corals that were exposed to both short- and long-term stress in 2022 (not conducted for parents in 2018 due to logistical constraints). (2) To corroborate the genetic correlation estimate from the animal model, we also tested the influence of parental short-stress tolerance (parent midpoint $DHW_{50\text{-short}}$ z-scores) on offspring long-stress tolerance (F1 $DHW_{50\text{-long}}$ z-scores) using Bayesian regressions, as described above for the parent-offspring regressions, except with a different construction of weakly informative priors. Instead, the weakly informative $\beta_1$ prior used here was such that values less than -2 or greater than 2 (i.e., a ± 1 standard deviation change in parental short-stress tolerance yielding a change in offspring long stress tolerance of ±2 standard deviations) were highly unlikely. This was achieved by setting the prior mean to 0 and the prior σ to 1.02 (i.e., precision of $1/1.02^2$). The influence of this weakly informative prior was tested by rerunning the regression using default non-informative $\beta_1$ priors (prior mean and precision set to 0 and 0.001, respectively) (Supplementary Fig. 9, equivalent result to Fig. 4c). (3) We also tested whether selectively breeding for short-stress tolerance provided offspring with long-stress tolerance, by comparing F1 $DHW_{50\text{-long}}$ between crosses with LL vs. HH parental heat tolerance$_{short}$ using LMMs, as described above. (4) Lastly, we tested the effect of parental heat tolerance$_{short}$ (LL vs. HH) on the progression of BSI responses throughout the heat stress exposure using GLMMs, as described above.

### Reporting summary

Further information on research design is available in the Nature Portfolio Reporting Summary linked to this article.

## Data availability

The data generated in this study on coral heat tolerance phenotypes, crosses conducted, coral colony sizes, symbiont ITS2 profiles, and temperature experiment conditions have been deposited at https://doi.org/10.25405/data.ncl.22812194. Source data are provided with this paper. All datasets analysed are publicly available as of the date of publication. ITS2 sequences have been archived publicly at NCBI under BioProject 864615 (accession code PRJNA864615). Source data are provided with this paper.

## Code availability

All datasets generated in this study and original R scripts used for analysis have been deposited at https://doi.org/10.25405/data.ncl.22812194. Processed symbiont community composition can be explored publicly at https://symportal.org.

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

## Acknowledgements

This works was supported by a European Research Council Horizon 2020 project CORALASSIST (725848) to JRG, and a Natural Environment Research Council ONE Planet Doctoral Training Partnership Studentship (NE/S007512/1) to LL. We also thank Prof. Loeske Kruuk and Dr. Albert Phillimore for helpful discussions on quantitative genetics, PICRC staff for supporting the research, Arius Merep for help with aquarium tank design and construction, and the boat operators Geory Mereb and Nelson Masang.

## Author contributions

Conceptualisation: A.H., L.L., E.B., J.B., J.R.K.C., A.J.E., Y.G., H.M.M., P.P., E.vd.S., M.S., A.J.W., and J.R.G. Data curation: A.H., L.L., A.J.E., E.B., J.B., and A.J.W. Formal analysis: L.L. and A.J. Funding acquisition: J.B., A.J.E., Y.G., and J.R.G. Investigation: A.H., L.L., E.B., L.B., D.B., J.B., J.R.K.C., R.dl.T.C., A.J.E., H.M.M., P.P., E.vd.S., M.S., A.W., and J.R.G. Methodology: A.H., L.L., E.B., L.B., D.B., J.B., J.R.K.C., R.dl.T.C., A.J.E., H.M.M., P.P., E.vd.S., M.S., A.W., A.J.W., and J.R.G. Project administration: A.H., L.L., and J.R.G. Software: A.H. and L.L. Visualisation: A.H. and L.L. Writing – original draft: A.H. and L.L. Writing – review & editing: All authors commented on the manuscript.

## Competing interests

The authors declare no competing interests.
