## [Peer Review File · Nature Communications]

Selective breeding enhances coral heat toleranceThis manuscript has been previously reviewed at another journal that is not operating a transparent peer review scheme. This document only contains reviewer comments and rebuttal letters for versions considered at *Nature Communications*.

REVIEWERS' COMMENTS

Reviewer #5 (Remarks to the Author):

My assigned task was to check if the authors have appropriately addressed the concerns of the reviewers, in particular reviewer #4.

From my perspective, the nature of the reviewer's comments have all been thorough, fair, constructive and added a lot of value to the study. I agree with the general sentiment of the reviewers, that the paper is a bit shy in taking a stance to whether selective breeding is a viable tool for coral reef restoration. Some of the reviewers also think that the main message is not a reflection of the results, and that the paper might be improved by a different style of communication and by including recommendations to the readers about how to actually perform selected breeding for corals.

At the same time, from my perspective, the authors have addressed the comments adequately and in detail.

However, the dispute remains with regards to the main message of the article. The authors state as their main message: "Selective breeding enhances coral heat tolerance". In contrary, the reviewers would like to see a discussion and interpretation of the results beyond the immediate results.

Essentially, this requested discussion would lead the paper to discuss questions, such as: is selective breeding of corals effective enough in order to boost coral thermal tolerance? What other measures are required? How can we improve the practise of coral selective breeding? etc.. These are of course big and delicate topics. There are multi million dollar project behind these project and opinionated people. I can understand that the authors from did cut the discussion short in order to leave it up to the reader to interpret this and read between the lines.

If I can add my 2 cents to this: Papers that are published in Nature comms can allow themselves to take a stance due to the impact of the journal and the potential lead role that they are taking directing the field in a certain direction. At the same time, I think that it is important to not overstate or understate the potential outcomes of selective breeding programs, especially in a higher impact paper like this that might be published in a high impact journal.

From my view, the field would benefit from an open discussion. The authors might do themselves a favour if they critically review their results and be a bit controversial, and rattle up the field. It seems like a healthy development to release a paper that discusses the potential success of selected breeding in an honest way, instead of pushing the narrative that the methods work and that we are heading in the right direction (that is of course not exactly what the paper is saying). Taking a stance, being more explicit and adding a more discussion in the line of what the authors request could open up exciting discussions.

There are quite a few papers out there saying that "Selective breeding enhances coral heat tolerance", but there is not a single one that puts this potential intervention technique into direct perspective of ecological relevance and speed of climate change.

I don't think that rejecting the paper would make any sense. In case of a rejection, those constructive reviews and the associated helpful discussions are not as present, they may get lost with different reviewers. From my perspective, it makes more sense to solve this with Nature comms.

So, in a nutshell, I think that the authors have sufficiently addressed the comments from all reviewers. This is from a perspective, that the paper is dealing with "fundamental science" and that anything else is beyond the scope; and that is totally fine. Perhaps, the editors of Nature comms need to decide if a paper with "fundamental science" is suitable for their journal, or if they are after a paper that goes beyond that. This might not be a decision that is only associated with this current article, but probably a general discussion about scope, potential citations and impact of the papers that are published in Nature comms.

Just one more thing:

I would appreciate, a stronger language around the importance of climate change. At the moment the main point of the paper reads like this:

"Our finding on the heritability of coral heat tolerance indicates that selective breeding could be a viable tool to improve population resilience. Yet, the moderate levels of enhancement we found suggest that the effectiveness of such interventions also depends on urgent climate action."

I would prefer a more direct language that they don't "depend" on urgent climate change action, but "demand" urgent climate change action. Having said this, the message around climate change action is so important that it deserves a seat in the front row in the paper. At the moment, this message is more conclusive and left in the background.

We thank the anonymous reviewer for their engagement with our research and for providing valuable and constructive comments. Overall, we have reworked the last paragraph of the discussion of the paper to address reviewer's concern, we have expanded and provided an open discussion on the potential success of selecting breeding corals for heat tolerance from an ecological perspective and in the context of the current rate of climate change. In this response to reviewers document, please note that all author responses are shown in **blue**. Quotes from the manuscript are shown in *italics*. Line numbers correspond to the 'simple view' of the revised manuscript (i.e., without showing tracked changes), and revisions/additions compared to the last draft are shown as *underlined italics*.

Reviewer #5 (Remarks to the Author):

My assigned task was to check if the authors have appropriately addressed the concerns of the reviewers, in particular reviewer #4.

From my perspective, the nature of the reviewer's comments have all been thorough, fair, constructive and added a lot of value to the study. I agree with the general sentiment of the reviewers, that the paper is a bit shy in taking a stance to whether selective breeding is a viable tool for coral reef restoration. Some of the reviewers also think that the main message is not a reflection of the results, and that the paper might be improved by a different style of communication and by including recommendations to the readers about how to actually perform selected breeding for corals.

At the same time, from my perspective, the authors have addressed the comments adequately and in detail.

Response 1: We appreciate the reviewer's thoughtful and positive feedback, as well as their thorough review of our responses to the other four reviewers.

However, the dispute remains with regards to the main message of the article. The authors state as their main message: "Selective breeding enhances coral heat tolerance". In contrary, the reviewers would like to see a discussion and interpretation of the results beyond the immediate results.

Essentially, this requested discussion would lead the paper to discuss questions, such as: is selective breeding of corals effective enough in order to boost coral thermal tolerance? What other measures are required? How can we improve the practise of coral selective breeding? etc.. These are of course big and delicate topics. There are multi-million dollar project behind these project and opinionated people. I can understand that the authors from did cut the discussion short in order to leave it up to the reader to interpret this and read between the lines.

Response 2: We agree with the reviewer that these are big and delicate topics that were outside the scope of our study. Further research is needed to estimate how effective selective breeding can be at enhancing the heat tolerance of coral populations and what the best practices are to do so. Our study demonstrates that selective breeding for heat tolerance in corals is feasible and that enhancements can be achieved in one generation, which is the first step needed to design experiments to answer other critical questions. We included some additional expansion of these ideas in the final paragraph, where we contextualise our results regarding the ecology

of reef systems, the rate of ocean warming, and the practicalities of what needs to be achieved for selective breeding to have a chance at improving coral population futures.

If I can add my 2 cents to this: Papers that are published in Nature comms can allow themselves to take a stance due to the impact of the journal and the potential lead role that they are taking directing the field in a certain direction. At the same time, I think that it is important to not overstate or understate the potential outcomes of selective breeding programs, especially in a higher impact paper like this that might be published in a high impact journal.

From my view, the field would benefit from an open discussion. The authors might do themselves a favour if they critically review their results and be a bit controversial, and rattle up the field. It seems like a healthy development to release a paper that discusses the potential success of selected breeding in an honest way, instead of pushing the narrative that the methods work and that we are heading in the right direction (that is of course not exactly what the paper is saying). Taking a stance, being more explicit and adding a more discussion in the line of what the authors request could open up exciting discussions.

There are quite a few papers out there saying that "Selective breeding enhances coral heat tolerance", but there is not a single one that puts this potential intervention technique into direct perspective of ecological relevance and speed of climate change.

Response 3: We thank the reviewer for this great suggestion. We have added an open discussion in the last paragraph of the Results and Discussion section (in the final Implications sub-section). Here we briefly discuss the potential of selecting breeding corals for heat tolerance from an ecological perspective and in the context of the current rate of climate change. In the limited space available, we make recommendations based on our results as follows:

L303-316: The proposition of using assisted evolution interventions to bolster reef resilience in the face of climate change is still highly controversial. It remains unknown whether such novel approaches will provide sufficient ecological benefits to warrant the expense and effort required to implement them. The results of our research offer reasons for both hope and caution simultaneously. On one hand, we found heat tolerance to be sufficiently heritable to suggest selective breeding could lead to significant enhancements in offspring population heat tolerance. On the other, the response to selection was relatively modest compared to projected ocean warming in the coming decades. In practice, to achieve significant trait enhancements that can keep pace with climate change will require ongoing rounds of selection over multiple generations – as is a standard approach for selective breeding programmes in other organisms⁵³ – and will need us to overcome several remaining challenges regarding how to incorporate selective breeding into reef rehabilitation programmes. These include targeting the appropriate species to benefit broader reef communities, upscaling selective breeding and out-planting efforts, overcoming post-deployment survival bottlenecks, and integrating with stakeholder and governance frameworks whilst securing sustainable financing models. In summary, our results show sufficient promise to warrant further research and development into selective breeding as an assisted evolution tool. However, we urge caution using these approaches until we have a better understanding of the relative benefits and risks. Without question, such efforts are only going to be worthwhile if in the long term we secure a future for coral reefs by rapidly reducing global carbon emissions and managing local-scale human

disturbances. These remain the greatest priorities for coral reefs and the people who depend on them.

I don't think that rejecting the paper would make any sense. In case of a rejection, those constructive reviews and the associated helpful discussions are not as present, they may get lost with different reviewers. From my perspective, it makes more sense to solve this with Nature comms.

So, in a nutshell, I think that the authors have sufficiently addressed the comments from all reviewers. This is from a perspective, that the paper is dealing with "fundamental science" and that anything else is beyond the scope; and that is totally fine. Perhaps, the editors of Nature comms need to decide if a paper with "fundamental science" is suitable for their journal, or if they are after a paper that goes beyond that. This might not be a decision that is only associated with this current article, but probably a general discussion about scope, potential citations and impact of the papers that are published in Nature comms.

Just one more thing:

I would appreciate, a stronger language around the importance of climate change. At the moment the main point of the paper reads like this:

"Our finding on the heritability of coral heat tolerance indicates that selective breeding could be a viable tool to improve population resilience. Yet, the moderate levels of enhancement we found suggest that the effectiveness of such interventions also depends on urgent climate action."

I would prefer a more direct language that they don't "depend" on urgent climate change action, but "demand" urgent climate change action. Having said this, the message around climate change action is so important that it deserves a seat in the front row in the paper. At the moment, this message is more conclusive and left in the background.

Response 4: Our edits have been done accordingly in the abstract.

L55-56: *Yet, the moderate levels of enhancement we found suggest that the effectiveness of such interventions also demands urgent climate action.*